# SwiftCAD: Efficient Parametric CAD Generation with Shared Decoder Transformers

Juyoung Kim[1*]    Seongjun Choi[2*]    Jiyeon Lim[3]    Soomok Lee[4†]

[1]Metacle    [2]Yonsei University    [3]Samsung Electronics    [4]Kennesaw State University

[1]jk042386@gmail.com, [2]sjchoi.dp@yonsei.ac.kr

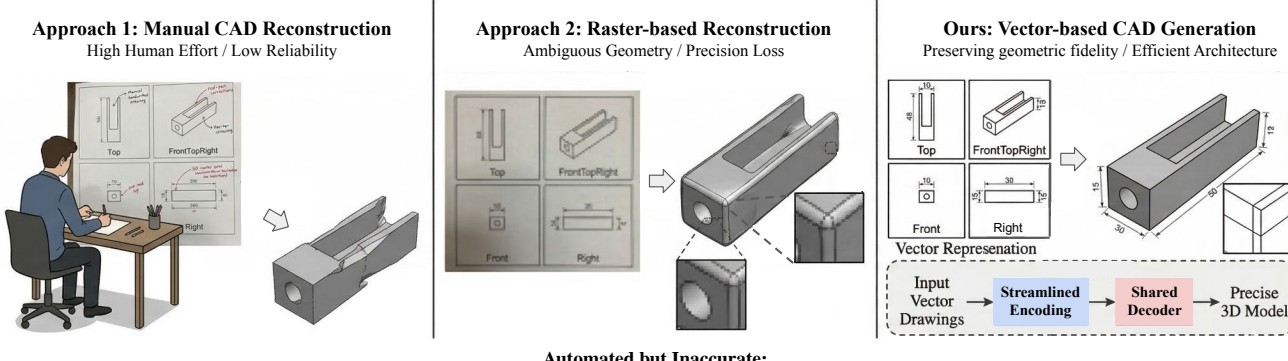

**Approach 1: Manual CAD Reconstruction**
High Human Effort / Low Reliability

**High Human Effort & Time-Consuming**

**Approach 2: Raster-based Reconstruction**
Ambiguous Geometry / Precision Loss

**Automated but Inaccurate:**
Pixel-level ambiguity, Quantization artifacts

**Ours: Vector-based CAD Generation**
Preserving geometric fidelity / Efficient Architecture

Vector Represenation

| Input Vector Drawings | → | Streamlined Encoding | → | Shared Decoder | → | Precise 3D Model |

**Automated, Accurate, and Efficient**

Figure 1. Comparison of CAD reconstruction paradigms. Manual reconstruction (left) requires extensive human effort and is prone to inconsistency, while raster-based approaches (middle) suffer from geometric ambiguity and precision loss due to pixel discretization and quantization artifacts. In contrast, our method (right) leverages vector-based representations together with a streamlined encoding pipeline and a weight-shared decoder, enabling accurate geometry reconstruction with significantly improved efficiency.

## Abstract

*Maintaining precise geometric dimensions is critical in Computer-Aided Design (CAD). Recent works have utilized representations for 3D generation, such as voxels, point clouds, and polygon meshes. In parametric CAD generation, approaches using Scalable Vector Graphics (SVG) have demonstrated superior performance in preserving dimensions. However, industrial applications require efficient operation on limited hardware, while existing architectures incur computational costs due to redundant structural components. In this paper, we propose two methods to improve efficiency. First, we remove redundant MLP layers to simplify the encoding process. Second, we adopt a weight-shared single-decoder (shared decoder) to jointly predict commands and parameters. To train and evaluate our methods, we use the CAD-VGDrawing dataset[30]. Our method achieves comparable generation accuracy while reducing model size and inference time. Project page: https://jadekim042386.github.io/SwiftCAD/*

## 1. Introduction

Computer-Aided Design (CAD) is a fundamental technology in modern manufacturing, enabling the precise design of products in domains such as aerospace, automotive, and medical devices. However, constructing CAD models manually is often labor-intensive and time-consuming, motivating increasing research interest in automated CAD generation. A key direction in this field is the generation of parametric CAD programs, where models are represented as sequences of design commands and geometric parameters. Such representations explicitly encode design intent and allow downstream modification through parameter editing. Recent works have explored various input modalities and representations, including command-parameter sequences and Boundary Representation (B-rep). Despite these advances, industrial CAD workflows impose strict requirements on geometric precision. Even small numerical inconsistencies can propagate through assemblies, which may affect downstream manufacturability and design consistency or manufacturing defects. As illustrated in Fig. 1, existing CAD reconstruction approaches exhibit a fundamental trade-off. Manual reconstruction requires significant human

---
*Equal contribution.

†Corresponding author.

effort and suffers from low reliability due to manual alignment. In contrast, raster-based automated methods reduce human effort but introduce geometric ambiguity and precision loss due to pixel-level discretization and quantization artifacts. These limitations make it challenging to simultaneously achieve automation, precision, and efficiency.

To address precision limitations in raster-based inputs, Drawing2CAD [30] introduced a framework that generates CAD sequences directly from Scalable Vector Graphics (SVG) drawings. By operating on vector primitives, this approach preserves geometric fidelity and achieves strong performance in CAD sequence generation. However, the architecture relies on a dual-decoder design that independently predicts command types and geometric parameters. While this decomposition simplifies the learning problem, it significantly increases model complexity and parameter count, which may hinder practical deployment and computational efficiency. Despite significant progress in representation learning and geometric fidelity, existing approaches often achieve these improvements by introducing increasingly complex model architectures. While such designs enhance modeling flexibility, they also lead to higher parameter counts and reduced efficiency, which can limit practical deployment in real-world CAD systems.

Moreover, CAD generation models are increasingly deployed in resource-constrained environments such as edge devices and interactive design tools, where inference latency and memory footprint are tightly budgeted [12]. Despite this, most prior works have focused primarily on improving generation quality, with relatively little attention given to systematically reducing model complexity. The efficiency–accuracy trade-off in parametric CAD generation remains largely unexplored.

This observation raises a fundamental question: is such architectural complexity truly necessary for accurate parametric CAD sequence generation? In this work, we revisit this architecture and identify key sources of structural redundancy, including the dual-decoder formulation and the embedding-stage MLP layers. Building on this analysis, we propose a streamlined architecture that significantly reduces model complexity while preserving geometric reasoning capability. First, we introduce a shared decoder that jointly models command and parameter generation while maintaining task-specific prediction heads. This design enables the model to capture correlations between commands and parameters while substantially reducing the total number of parameters. Second, we simplify the encoding pipeline by removing the embedding-stage MLP layer, allowing the Transformer encoder to directly process embedded tokens. Empirical analysis shows that the self-attention mechanism alone is sufficient to capture the required cross-field interactions. Through extensive experiments on the CAD-VGDrawing dataset[30], we demonstrate that the proposed

architecture achieves up to a 64.74% reduction in both parameter count and model size compared to the baseline, while maintaining competitive generation accuracy within 0.5% in command accuracy and 0.9% in parameter accuracy. These results suggest that much of the architectural complexity in existing CAD generation models is unnecessary, and that carefully designed lightweight architectures can match the performance of significantly larger models. The main contributions of this work are summarized as follows:

- We analyze the architectural design of Drawing2CAD [30] and identify structural redundancies in both the dual-decoder structure and the embedding-stage MLP layer.
- We propose a shared decoder architecture that jointly models command and parameter generation, achieving a 23.61% reduction in parameters with virtually no loss in accuracy, and up to 64.74% when combined with encoding simplification.
- We introduce a streamlined encoding pipeline that removes the embedding-stage MLP layer, demonstrating that Transformer self-attention alone can effectively capture cross-field interactions for CAD sequence generation.

## 2. Related Work

### 2.1. CAD Program Generation

Generating parametric CAD models as sequences of commands and parameters has become a dominant paradigm, enabled by large-scale CAD repositories. Earlier work such as 3D-PRNN [41] demonstrated autoregressive generation of 3D shape primitives, and large-scale datasets such as FloorPlanCAD [7] have further supported research on vector-based CAD understanding. The ABC dataset [18] provides a large collection of CAD geometries, while Deep-CAD [38] extracts structured modeling sequences from Onshape design histories [40]. This representation explicitly captures design intent and supports downstream editability. Subsequent works have focused on improving controllability and diversity. SkexGen [39] introduces disentangled latent representations for sketch topology and extrusion parameters, improving controllability but requiring complex latent design. Fusion360 Gallery [37] provides real-world CAD programs for data-driven modeling. CS-GNet [32] and UCSG-Net [16] model constructive solid geometry programs, enabling explicit boolean operations but limiting flexibility in free-form design. NeuralCAD [6] and CADNet [36] further explore sequence-based CAD generation with improved structural consistency, though often at increased model complexity. Recent approaches such as CADTransformer [8] and PolyGen [26] extend sequence modeling to more complex geometric structures, improving expressiveness while introducing additional computational

overhead.

Sketch-based interfaces have also been widely explored. Sketch2CAD [20], Free2CAD [21], and DeepSVG [1] treat sketch understanding as a sequence translation problem, bridging human input and executable CAD programs. While these approaches improve usability, they primarily focus on input modality rather than architectural efficiency.

Overall, existing CAD program generation methods emphasize expressiveness and controllability, often overlooking model efficiency and scalability, which are critical for practical deployment.

## 2.2. Input Representations and Geometric Fidelity

Preserving geometric precision is a fundamental challenge in CAD generation. Raster-based methods, such as Pixel2Mesh [34] and IM-NET [4], suffer from discretization artifacts that degrade geometric accuracy. Implicit representations, including Occupancy Networks [25] and DeepSDF [28], improve surface reconstruction quality but do not explicitly encode parametric design intent.

Vector-based representations provide a more precise alternative. DeepSVG [1] learns hierarchical representations of SVG primitives, while Im2Vec [31] and DiffVG [22] enable differentiable vector graphics rendering. Drawing2CAD [30] leverages SVG primitives to directly generate CAD programs, preserving geometric fidelity while enabling learning-based modeling. Similarly, SVG-VAE [24] and VectorFusion [15] explore generative modeling in vector space with improved structural consistency.

These works demonstrate that representation choice plays a crucial role in balancing geometric precision and learning capability. However, they do not address architectural redundancy in downstream CAD generation models.

## 2.3. Architectures for CAD Generation

Transformer-based architectures have become the standard for CAD sequence modeling. DeepCAD [38] formulates CAD generation as a sequence-to-sequence problem using Transformers, demonstrating strong capability in modeling long-range dependencies. Drawing2CAD [30] extends this framework with a dual-decoder architecture for command and parameter prediction, simplifying task decomposition but increasing parameter count.

Recent works explore more expressive generative paradigms. VQ-CAD [33] incorporates vector quantization within a diffusion framework for 3D CAD synthesis, achieving high-quality generation at the cost of increased complexity. Diffusion-based methods such as Point-E [27] and ShapeCrafter [10] demonstrate strong generative performance for 3D structures but lack explicit parametric representations. In parallel, large pretrained models have been applied to CAD generation. TCADGen [23] integrates visual and textual modalities, while CAD-Coder [13] lever-

ages vision–language models to generate executable CAD programs, improving generalization but requiring large-scale training. Graph-based approaches such as Mesh-GraphNet [29] and Neural B-Rep [14] explicitly model geometric relationships, while implicit neural representations [25, 28] capture continuous geometry. Although these approaches enhance modeling flexibility, they typically rely on complex architectures and large parameter counts.

Overall, improvements in modeling capability are often achieved by increasing architectural complexity, raising concerns about efficiency and scalability.

## 2.4. Architectural Efficiency in CAD Models

Despite rapid progress in modeling capability, architectural efficiency remains underexplored in CAD generation. Many state-of-the-art approaches [8, 13, 23, 33] rely on multiple Transformer decoders, hierarchical codebooks, or additional MLP layers, leading to increased parameter counts and inference latency.

Efficiency has been studied in other domains through weight sharing and parameter-efficient Transformer designs, such as ALBERT [19], Linformer [35], and Performer [5], which reduce redundancy in attention mechanisms. Similarly, auto-scaling strategies for Vision Transformers [2, 3] and quantization-based compression methods [11, 17] have shown that significant parameter reduction is achievable with minimal performance loss. In the 3D domain, LightGaussian [9] demonstrates effective compression of 3D Gaussian representations through pruning and distillation. However, these principles have not been systematically applied to structured CAD sequence generation.

In this work, we revisit the architectural design of Drawing2CAD [30] and identify key sources of redundancy, including the dual-decoder structure and embedding-stage MLP layers. We show that much of this architectural complexity is unnecessary, and that comparable or better performance can be achieved with a significantly simpler and more efficient design. By introducing a shared decoder and a streamlined encoding pipeline, our approach reduces parameter count and inference time while preserving geometric reasoning capability, making CAD generation more practical for real-world deployment.

## 3. Preliminary: Drawing2CAD

A CAD operation sequence is a textual representation of computer-aided design, comprising commands and parameters. Our research focuses on the generation of single objects using a specific subset of CAD commands: "Line", "Circle", "Arc", and "Extrude". Given an input SVG sequence, the goal is to generate a corresponding CAD operation sequence that accurately reconstructs the underlying 3D geometry. Similar to the CAD sequence, an SVG

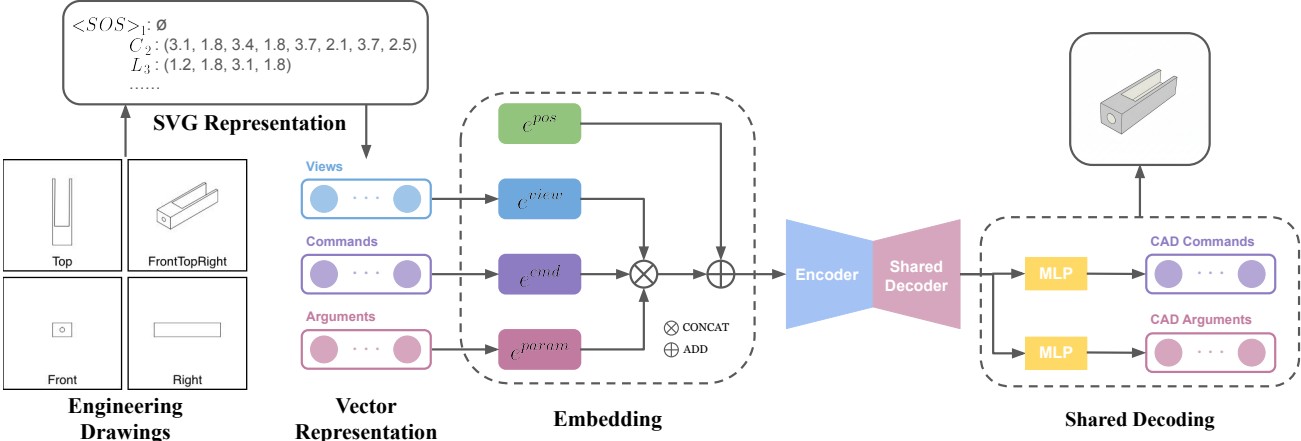

Figure 2. Overview of the proposed CAD sequence generation pipeline. Engineering drawings are decomposed into view, command, and parameter fields, which are embedded and concatenated to form the encoder input. The encoded representation is processed by a weight-shared Transformer decoder with task-specific heads for CAD command and parameter prediction, producing a structured CAD sequence.

drawing sequence serves as a textual representation for Scalable Vector Graphics. In this work, we simplify the SVG sequence into two primary SVG commands: "LineTo" for straight lines and "Cubic Bezier" for smooth contours. Throughout this paper, we refer to input-side drawing instructions (e.g., LineTo, CubicBezier) as SVG commands, and output-side modeling operations (e.g., Line, Circle, Arc, Extrude) as CAD commands. In the encoder, SVG commands are tokenized into a compact set of command tokens (e.g., ⟨SOS⟩, L, C, ⟨EOS⟩). A key characteristic of both sequence types is that they facilitate design modifications through parameter adjustment, thereby enabling automatic model updates. Further technical specifications are detailed in DeepCAD [38] and DeepSVG [1].

**Embedding.** View labels and command tokens are represented as 4-dimensional one-hot vectors and embedded. Arguments are quantized into 8-bit integers and represented as 257-dimensional one-hot vectors before embedding, then flattened with a linear projection for inter-argument interaction.

**Transformer.** Drawing2CAD [30] consists of a Transformer encoder and a dual-decoder. In the encoder, individual embeddings are concatenated, and a linear transformation (MLP) is applied to enable cross-field interaction. After positional encoding, the sequence is processed by the Transformer encoder to produce a 256-dimensional latent representation $z$ that encodes global geometric and structural information.

The dual-decoder consists of independent Transformer decoders that predict CAD commands and arguments. Both decoders receive the latent code as a constant embedding input. Each output is projected through separate linear layers to obtain the predicted command and argument. For argument generation, command-guided generation is applied by

incorporating the output of the command decoder into the argument decoder.

While this design effectively captures the structure of CAD programs, it introduces additional architectural complexity due to the use of separate decoders and embedding-stage MLP layers. This motivates the need for a more efficient formulation that can maintain geometric reasoning capability while reducing redundancy.

## 4. Method

**Problem Formulation.** Given an input SVG sequence $X = x_1, x_2, \ldots, x_n$, our goal is to learn a mapping function $f_\theta$ that generates a corresponding CAD operation sequence $Y = y_1, y_2, \ldots, y_m$, where each element consists of a CAD command type and its associated geometric parameters. The objective is to accurately reconstruct the underlying 3D geometry while preserving the structural dependencies between commands and parameters. This formulation follows the sequence-to-sequence paradigm while explicitly modeling structured outputs.

Building on the limitations identified in the baseline architecture, we propose a streamlined design that removes redundant components while preserving modeling capacity. Specifically, we eliminate the embedding-stage MLP and introduce a shared decoder to jointly model command and parameter generation. These modifications are designed to reduce architectural complexity while maintaining the ability to capture geometric relationships.

As illustrated in Fig. 2, the input drawing is decomposed into view types, command tokens, and geometric parameters. These components are embedded into a unified representation and processed by a Transformer encoder–decoder framework. The encoder captures global contextual relationships across views, while the decoder autoregressively

generates structured CAD sequences.

## 4.1. Streamlined Encoding without MLP

The original framework applies an additional MLP after the embedding stage to facilitate cross-field interactions between view, command, and parameter tokens. However, we observe that the categorical cardinality of these fields is extremely low: the view field contains only four types (top, front, right, isometric), and the command token field also consists of a small set of tokens ($\langle SOS \rangle$, $L$, $C$, $\langle EOS \rangle$). This suggests that complex linear transformations for feature interaction may be unnecessary and could introduce redundant parameters.

Motivated by this observation, we remove the embedding-stage MLP and directly feed the concatenated embeddings into the Transformer encoder. This allows the model to rely on self-attention to capture both internal dependencies within each field and interactions between different fields. The input sequence is defined as:

$$E_{input} = \text{CONCAT}(e_{\text{view}}, e_{\text{cmd}}, e_{\text{param}}) + e_{\text{pos}}. \quad (1)$$

The view and command embeddings are obtained as:

$$e_{\text{view}} = W_{\text{view}} \delta_i^v, \quad (2)$$
$$e_{\text{cmd}} = W_{\text{cmd}} \delta_i^c, \quad (3)$$

where $W_{\text{view}}, W_{\text{cmd}} \in \mathbb{R}^{d_e \times 4}$ are learnable embedding matrices.

For geometric parameters, each command is associated with eight parameters, each quantized into 8-bit integers and represented as 257-dimensional one-hot vectors. The parameter embedding is computed as:

$$e_{\text{param}} = W_{\text{param}}^a; \text{flatten}(W_{\text{param}}^b \delta_i^p), \quad (4)$$

where $W_{\text{param}}^b \in \mathbb{R}^{d_e \times 257}$ embeds each parameter independently, and $W_{\text{param}}^a \in \mathbb{R}^{d_e \times (8d_e)}$ models inter-parameter interactions.

By removing the additional MLP layers, the encoder maintains its ability to capture global dependencies through self-attention while significantly reducing parameter count. In practice, this simplification does not degrade representation quality, as demonstrated in our experiments.

## 4.2. Shared Decoder Architecture

In CAD sequence generation, command types and their associated parameters are inherently correlated. The dual-decoder design in the baseline explicitly separates these tasks, simplifying optimization but increasing model complexity. To address this inefficiency, we adopt a shared decoder architecture. Instead of using separate decoders, a single Transformer decoder jointly models both CAD command and parameter generation. This design allows

shared representations to implicitly encode dependencies between command types and their parameters. Formally, given the encoded representation $z$, the decoder produces hidden states in an autoregressive manner:

$$h_t = \text{Decoder}(z, y_{<t}), \quad (5)$$

where $y_{<t}$ denotes the sequence of previously generated tokens. This formulation enables the model to condition each prediction on both the global context and prior outputs. The hidden states are then passed to task-specific output heads:

$$\hat{c} * t = \text{MLP} * \text{cmd}(h_t), \quad (6)$$
$$\hat{p} * t = \text{MLP} * \text{param}(h_t), \quad (7)$$

where $\hat{c}_t$ and $\hat{p}_t$ denote the predicted command type and geometric parameters, respectively. By sharing the decoder, the model can jointly learn structural relationships without requiring explicit command-guided mechanisms.

## 4.3. Training Objective

We train the model using teacher forcing, where the ground-truth sequence is provided during training. The objective combines command classification and parameter prediction losses. The CAD command prediction loss is defined as:

$$\mathcal{L}_{\text{cmd}} = -\sum_t \log P(c_t \mid z, y * < t), \quad (8)$$

and the parameter prediction loss is given by:

$$\mathcal{L}_{\text{param}} = -\sum_t \log P(p_t \mid z, y * < t). \quad (9)$$

The overall objective is:

$$\mathcal{L} = \mathcal{L}_{\text{cmd}} + \lambda \mathcal{L}_{\text{param}}, \quad (10)$$

where $\lambda$ balances the contribution of parameter prediction. In practice, we set $\lambda = 1$. At inference time, the CAD sequence is generated autoregressively until an end-of-sequence token is produced.

Overall, the proposed architecture removes redundant components while preserving the ability to model structured geometric relationships, resulting in an efficient and effective CAD sequence generation framework.

## 5. Experiments

### 5.1. Implementation Details

We evaluate our proposed method using the CAD-VGDrawing dataset [30] created by Drawing2CAD. This dataset comprises vector drawings projected from 3D objects into four distinct views: top, front, right, and isometric (front-top-right). In the original study, three input configurations were explored: isometric only, orthographic

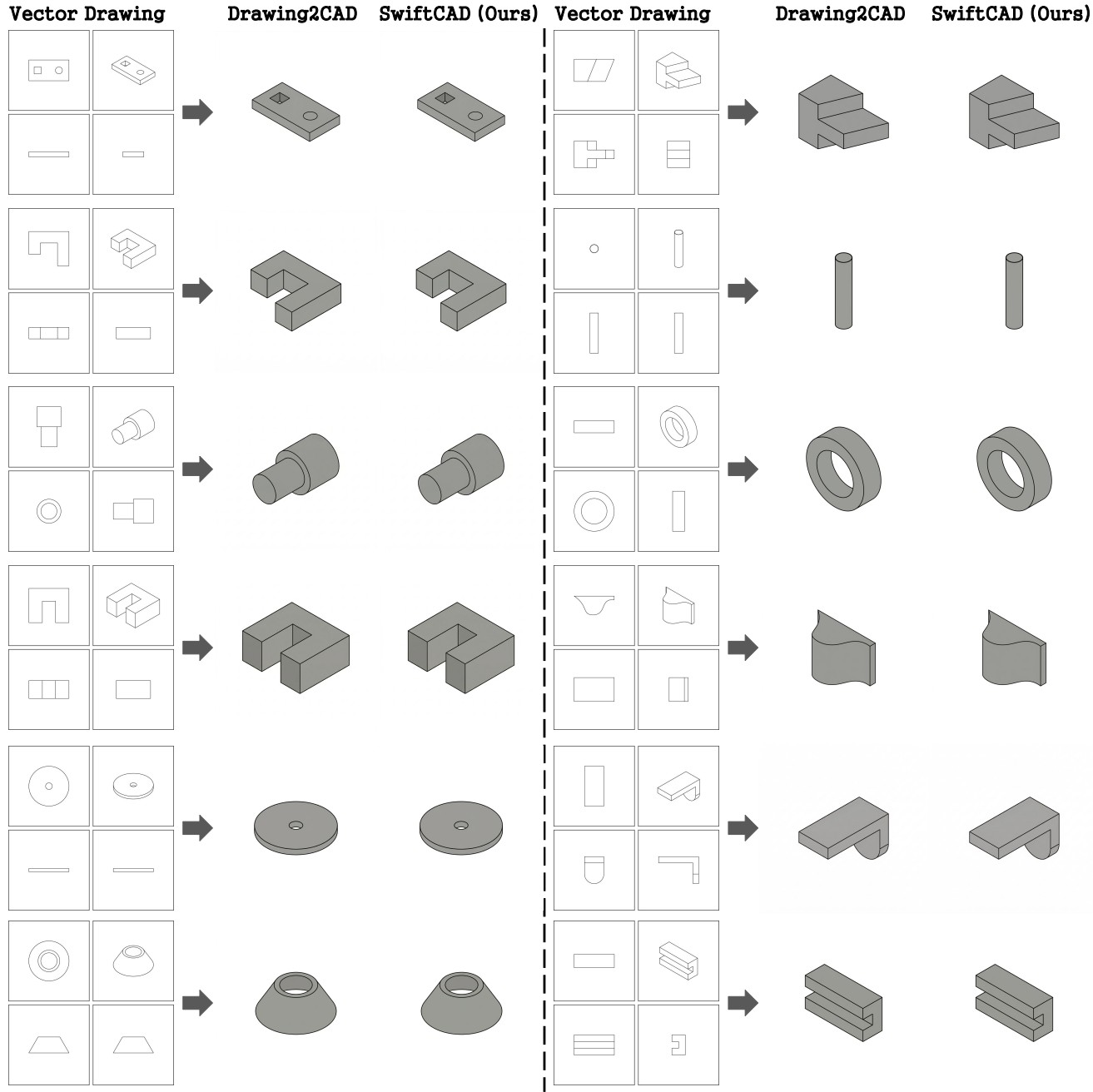

Figure 3. Qualitative comparison between the baseline and the proposed method (SwiftCAD). Our method produces geometrically consistent CAD reconstructions across diverse shapes, including planar, cylindrical, and composite structures. Despite significant reductions in complexity, the generated results remain visually comparable to the baseline, preserving both structural topology and geometric details.

(top, front, and right), and a combination of both isometric and orthographic views (4x). Given that the 4x views demonstrated the highest performance in the baseline study, we adopt it as our standard input format for all comparative evaluations. We evaluate our method using the same dataset with 4x views. Experiments are conducted on a single NVIDIA RTX 4090 GPU with a batch size of 256

over 200 epochs using the Adam optimizer (LR: 0.001) and a linear warm-up scheduler. Following Drawing2CAD [30], we evaluate generation quality using command accuracy ($ACC_{cmd}$), parameter accuracy ($ACC_{param}$), Invalidity Ratio (IR), and Mean Chamfer Distance (MCD). IR measures the proportion of invalid CAD sequences that fail geometric reconstruction, while MCD evaluates geo-

| Method | Parameters | $ACC_{cmd}$ | $ACC_{param}$ |
|---|---|---|---|
| w/o MLP (144) | 4,423,140 | 82.38 | 78.75 |
| w/o MLP (192) | 6,532,488 | 82.52 | 78.91 |
| w/o MLP (256) | 9,844,024 | 82.64 | 79.13 |

Table 1. Ablation study on removing the embedding-stage MLP with the dual-decoder architecture. The numbers in parentheses denote the embedding dimension $d_e$.

metric similarity between the generated and ground-truth CAD models.

## 5.2. Qualitative Results

We present qualitative comparisons in Fig. 3. Across all examples, our method generates CAD models that are visually indistinguishable from those produced by the baseline. Both basic primitives (planar extrusions, cylindrical structures) and complex compositions involving multiple extrusions are reconstructed with high accuracy, indicating that the shared decoder effectively captures command semantics and parameter relationships. Even for geometries requiring precise multi-view alignment (e.g., L-shaped structures and stepped solids), our method maintains consistent topology. Despite the substantial reduction in parameters, no significant degradation in visual quality is observed, confirming that a more compact design can achieve comparable results.

## 5.3. Quantitative Results

We quantitatively compare our method with the baseline Drawing2CAD [30], as summarized in Table 2. The test set contains 7,881 samples. All inference times are measured as the total wall-clock time over the entire test set with a batch size of 256.

Replacing the dual-decoder with a shared decoder reduces the number of parameters and model size by 23.61%, while maintaining nearly identical performance in both command and parameter accuracy. In addition, a slight improvement in inference time is observed. This result indicates that explicitly separating command and parameter generation is not necessary, and that a shared representation can effectively capture their dependencies.

Further simplification by removing the embedding-stage MLP leads to a substantial reduction in model complexity. In the most compact configuration ($d_e = 144$), the model achieves a 64.74% reduction in parameters along with the fastest inference time. Although a minor decrease in $ACC_{param}$ is observed, the overall performance remains comparable, suggesting that the contribution of the MLP layer to predictive accuracy is limited. A similar trend is observed in IR and MCD, where the compact models show only marginal degradation despite substantial reductions in model complexity.

To analyze the effect of model capacity, we vary the em-

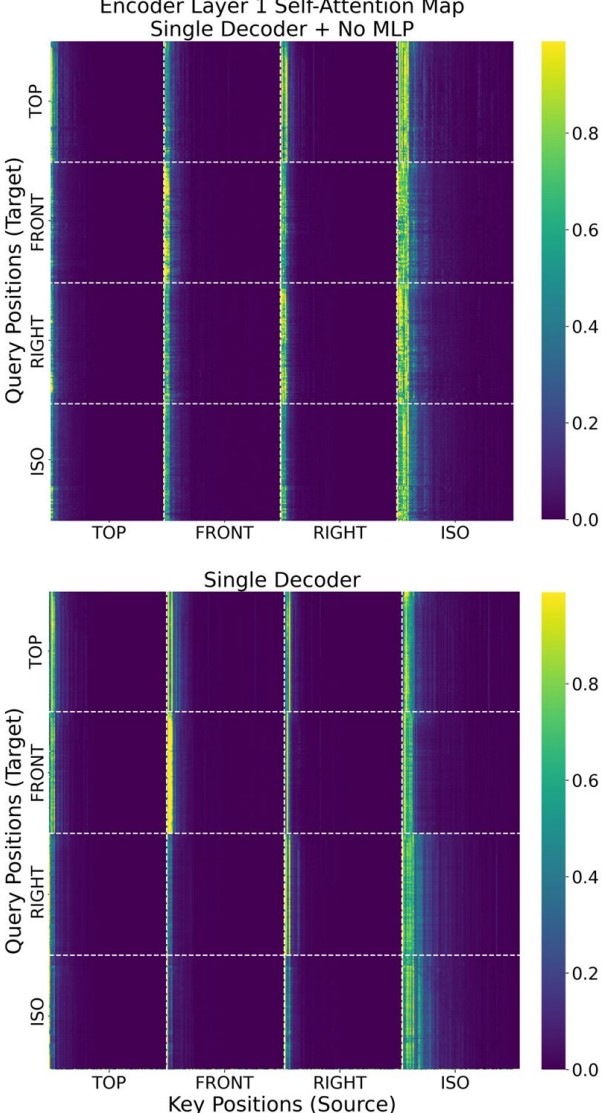

Figure 4. Comparison of encoder self-attention maps between the baseline (with MLP) and the proposed w/o MLP configuration. Both models exhibit highly similar structured attention patterns, including strong vertical alignments at view boundaries and consistent cross-view interactions.

bedding dimension ($d_e = 144, 192, 256$). As the embedding dimension increases, both parameter count and performance approach those of the baseline. Notably, at $d_e = 256$, the model achieves nearly identical accuracy to the baseline while retaining a simplified architecture. Interestingly, lower-dimensional configurations maintain competitive CAD command accuracy, indicating that CAD command prediction is less sensitive to embedding dimensionality.

Overall, these results demonstrate that the proposed architectural simplifications significantly improve efficiency while preserving generation performance, highlighting the

| Method | Parameters | File Size (MB) | Inf. Time (s) | $ACC_{cmd}\uparrow$ | $ACC_{param}\uparrow$ | $IR\downarrow$ | $MCD\downarrow$ |
|---|---|---|---|---|---|---|---|
| Baseline (Drawing2CAD [30]) | 10,109,894 | 38.57 | 8.4815 | 82.76 | 79.23 | 20.44 | 11.16 |
| Shared Decoder | 7,722,438 | 29.46 | 8.4329 | 82.42 | 79.13 | 21.52 | 11.29 |
| Shared + w/o MLP (256) | 7,745,850 | 29.55 | 8.3925 | 82.20 | 79.21 | 21.55 | 11.30 |
| Shared + w/o MLP (192) | 5,204,234 | 19.85 | 8.4447 | 82.17 | 78.87 | 21.75 | 11.66 |
| Shared + w/o MLP (144) | 3,564,806 | 13.60 | 7.7575 | 82.37 | 78.55 | 21.89 | 11.90 |

Table 2. Comparison of quantitative results between the baseline and our methods. IR and MCD denote Invalidity Ratio and Mean Chamfer Distance, respectively. Inference time denotes the total wall-clock time for processing the entire test set using a batch size of 256 on a single NVIDIA RTX 4090 GPU.

effectiveness of reducing structural redundancy in CAD sequence generation models.

### 5.4. Ablation Study

We conduct an ablation study to isolate the effect of each architectural modification. As reported in Table 1, removing the MLP alone (with the dual-decoder) yields consistent trends with the combined configuration (Table 2), confirming that the MLP contributes minimally to prediction accuracy regardless of decoder design.

**Effectiveness of Removing MLP layers.** To further validate this finding, we compare the encoder self-attention maps in Fig. 4. The baseline and w/o MLP configurations exhibit highly similar structured patterns, with strong vertical alignments at view boundaries and consistent cross-view interactions. Notably, the w/o MLP configuration produces even clearer attention patterns, indicating that the Transformer's self-attention mechanism alone is sufficient to model cross-field interactions without additional MLP-based transformations.

### 6. Conclusion

In this paper, we revisited the architectural design of parametric CAD generation and identified structural redundancies in both the encoding and decoding stages. We proposed a streamlined encoding pipeline without the embedding-stage MLP and a shared decoder architecture that jointly models command and parameter generation. Extensive experiments demonstrate that our approach significantly reduces model size and inference time while maintaining competitive generation performance. These results suggest that carefully designed lightweight architectures can effectively balance efficiency and generation quality in CAD sequence generation. We believe our approach provides a practical direction for lightweight CAD assistance and resource-constrained deployment scenarios.

### 7. Limitations and Future Work

Despite the efficiency gains achieved by our streamlined architecture, several limitations remain. First, our method is evaluated on a constrained set of CAD commands (e.g.,

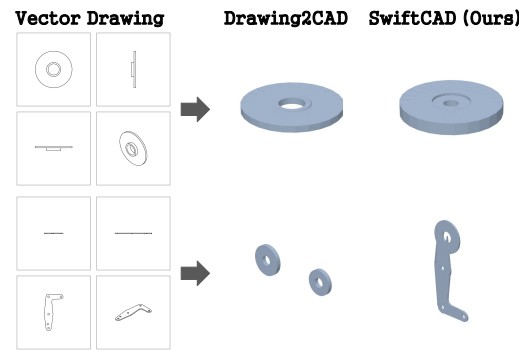

Figure 5. Qualitative analysis of representative failure cases. Reconstruction errors primarily occur in sequences exceeding approximately 80 tokens, where autoregressive error accumulation leads to geometric inconsistencies and topological errors, particularly in thin-walled and closely spaced structures.

Line, Circle, Arc, and Extrude), which may limit its applicability to more complex CAD modeling scenarios involving diverse operations and hierarchical dependencies. Second, while the proposed shared decoder architecture effectively captures the correlation between CAD commands and parameters, it relies on autoregressive generation, which may lead to error accumulation in long sequences. Incorporating more robust sequence modeling strategies or non-autoregressive alternatives could further improve stability. As illustrated in Fig. 5, we observe that reconstruction quality degrades notably when the target CAD sequence exceeds approximately 80 tokens, where autoregressive error accumulation leads to progressive parameter drift in later commands. In particular, failures tend to concentrate on shapes involving thin-walled features or closely spaced extrusions, where small positional errors in early commands propagate and compound through subsequent predictions. These observations suggest that sequence length, rather than geometric complexity alone, is a primary factor in reconstruction failure. Future work could explore improved parameter representations, continuous geometry modeling, or integration with large-scale pretraining.

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
