# OpenReview forum: "SwiftCAD: Efficient Parametric CAD Generation with Shared Decoder Transformers"
_thecvf.com/CVPR/2026/Workshop/3D4S — CVPR 2026 Workshop 3D4S Poster_

### Official Review · Reviewer_VMzw · 2026-04-19
**Useful Drawing2CAD simplification, but evaluation needs CAD-level metrics**

**Rating:** 6
**Confidence:** 4

**Review:**

I like this paper because the motivation is clear and I also find the paper relevant to 3D4S. The method is easy to understand, and the experiments show a clear reduction in parameter count and model size while keeping command/parameter accuracy close to the Drawing2CAD baseline. It's also helpful to see authors discuss failure cases for long sequences.

My main concern is that the metrics are useful for comparing against Drawing2CAD, but they are not really good enough to establish final CAD quality. Specifically, command accuracy and parameter accuracy are useful for checking whether the shared decoder affects token-level prediction, but the paper is missing other metrics like Invalidity Ratio and Mean Chamfer Distance. Since SwiftCAD claims comparable CAD generation quality, the authors should report these CAD-level metrics as well like Drawing2CAD paper. Otherwise, it is unclear whether similar token accuracy translates into equally valid and geometrically comparable CAD models.

Evaluating only on CAD-VGDrawing is understandable/acceptable given the task and its close connection to Drawing2CAD, but it limits how broadly the conclusions can be interpreted. It would be helpful to test whether the same simplifications hold for richer command sets, or more diverse CAD sources.

Lastly, since the reported accuracy differences are small, repeated runs or standard deviations would make the comparison more reliable.

[Minor] The ablation results are useful, but they are split across Tables 1 and 2. It would be helpful to combine them, so readers can more easily separate the effects of decoder sharing, MLP removal, and embedding dimension on both model size and accuracy.

---

### Official Review · Reviewer_FX9J · 2026-04-25
**SwiftCAD: A Practical but Incremental Efficiency-Oriented CAD Generation Method**

**Rating:** 5
**Confidence:** 4

**Review:**

## Strengths

1. Clear and practical motivation.
   The paper focuses on an important practical issue: efficient CAD generation. In real CAD workflows, model size, inference latency, and deployment cost matter, especially for interactive design tools or resource-limited devices. The motivation is reasonable and clearly presented.

2. Simple and effective architectural design.
   The proposed method is straightforward: remove unnecessary MLP layers and share the decoder between command and parameter prediction. Although simple, this design directly targets redundancy in Drawing2CAD and achieves meaningful parameter reduction.

3. Good efficiency improvement.
   The most compact version reduces parameters from 10.1M to 3.56M and file size from 38.57MB to 13.60MB. This is a substantial reduction. The shared-decoder-only variant also achieves almost the same accuracy as the baseline with fewer parameters, which supports the claim that the dual-decoder structure may be redundant.

## Weaknesses / Discussion

1. Limited novelty.
   The main idea is architectural simplification through decoder sharing and MLP removal. Weight sharing and removing redundant layers are common efficiency techniques in Transformer models. The contribution is useful, but it is more like an engineering optimization of Drawing2CAD than a fundamentally new CAD generation method. The originality is therefore moderate.

2. Baseline comparison is too narrow.
   The paper mainly compares against Drawing2CAD. Since the proposed method is explicitly a simplified Drawing2CAD, this comparison is necessary but not sufficient. The paper should also compare with other CAD generation or sketch-to-CAD methods, such as DeepCAD, SkexGen, CADTransformer, or more recent CAD sequence generation models, even if their input/output settings differ. Without broader comparisons, it is hard to evaluate the method’s position in the overall CAD generation literature.

3. Accuracy improvement is not significant.
   The paper’s main claim is efficiency, not quality. However, the generation accuracy is mostly comparable or slightly worse than the baseline. For example, the most compact version has parameter accuracy 78.55, lower than the baseline’s 79.23. This is acceptable for compression, but it means the paper does not demonstrate a stronger generative model.

---

### Official Review · Reviewer_Vqc4 · 2026-04-25
**Cutting parameters while maintaining precision**

**Rating:** 7
**Confidence:** 3

**Review:**

# Summary

 The paper studies parametric CAD program generation from vector drawings (SVG) and asks whether the architectural complexity found in prior models is necessary to preserve geometric fidelity. The main contribution is a streamlined Transformer encoder-decoder that reduces the parameter count and model size while maintaining similar accuracy to the baseline.
 The method takes in a SVG sequence and outputs CAD operation sequence (in the form of commands and arguments) to reconstruct the 3D geometry. In particular the method considers straight "line" and "cubic bezier" contours in the inputs and reconstructs with the  "Line", "Circle", "Arc", and "Extrude" output commands.

# Strengths

The paper clearly targets efficiency which is a deployment concern for CAD tools at an industrial scale or on memory-constrained devices and demonstrates simpler architectures that can match the performance of heavier baseline methods. Results shows substantial reduction in parameter count, file size and inference time.
The architectural changes are well-motivated with empirical results.

# Weaknesses

There is a important gap in the motivation of this paper and the results the method produces. The paper motivates applications in safety-critical domains like aerospace, automotive, and medical devices which require "precise design of products". Nonetheless, results show that this method reduces precision (albeit largely minimized compared to the efficiency gains). The tradeoff for precision for safety-critical applications, and efficiency improvements results in a disconnect in the motivation and results.  Clarifying the positioning, would strengthen the importance/utility of this work.

I am not an expert in CAD design but I would like to understand why the paper frames human efforts as "suffering from low reliability" and automated methods suffering from "geometric ambiguity and precision loss". Again there seems to be a motivation disconnect. For  safety-critical domains like aerospace, automotive, and medical devices, is there some methods that are *not* suffering from some type precision/reliability? If such a method exists would you consider this a baseline?

---

### Decision · Program_Chairs · 2026-04-28

Accept (Poster)